# A Selection Method for Restoration Mortars Using Sustainability and Compatibility Criteria

José Diaz-Basteris [1],*, Beatriz Menéndez [1],*, Javier Reyes [2] and Julio C. Sacramento Rivero [3]

1   Geosciences and Environment Cergy, CY Cergy Paris Université, 95000 Neuville sur Oise, France
2   LANCIC-CICORR, Universidad Autónoma de Campeche, Campeche 24000, Mexico
3   Faculty of Chemical Engineering, Universidad Autónoma de Yucatán, Mérida 97000, Mexico
*   Correspondence: jose.diaz-basteris@cyu.fr (J.D.-B.); beatriz.menendez@cyu.fr (B.M.);
    Tel.: +33-767-068-896 (J.D.-B.)

**Abstract:** This work proposes sustainability criteria for the selection or design of restoration mortars based on their physical and mechanical properties, durability, price in the French market, and the environmental impact estimated by the global warming potential. A score is assigned to the mortars based on normalized values of their physical and mechanical properties. A total of 24 formulations of restoration mortars were characterized, and their scores were compared. A case study showing the application of the proposed selection method is presented, focused on the restoration of historical monuments in Paris, France, built with Lutetian and Euville stones. In this case, hydraulic lime mortars were the most sustainable options. The application of the method is also projected for global application, as showcased for the restoration of Mayan stones in Southern Mexico.

**Keywords:** lime restoration mortars; global warming potential (environmental impact); hydraulic lime; durability; Lutetian stone; Euville stone; Maya stones





## 1. Introduction

The conceptual and methodological baggage of geosciences constitute cognitive tools that enrich the view of environmental problems. A deep understanding of the raw materials in geosciences help to understand the evolution of construction materials. Allowing to put into perspective and measure with greater precision the global impact of construction activities on the planet.

A mortar is a mixture of binders, aggregates, and additives used as a construction material to rig building elements, such as bricks, ashlars, masonry, or concrete blocks, to fill the spaces between the blocks, and for cladding [1]. The standard EN 16572 defines a mortar as a "material traditionally composed of one or more (usually inorganic) binders, aggregates, water, possible additives, and admixtures combined to form a paste used in masonry for bedding, jointing, bonding, and for surface finishing (plastering and rendering) of masonry units, which subsequently sets to form a stiff material" [2]. Restoration mortars are used to cover or replace lost parts in structures, monuments, or sculptures, mosaics, mural painting, or to make mockups of architectonic, archaeological, or sculptural elements of cultural value, etc. These mortars must be carefully selected before application to ensure their suitability to interact with the ancient substrate, allowing for water and air transport with physical and mechanical compatibility [3,4]. Some ancient mortars or ancient techniques are still being studied and different uses of mixtures of organic and inorganic materials are still being discovered. Ancient additives and admixtures show the advanced understanding of ancient cultures of the interaction of materials, especially pozzolanic reactions [5–8].

The concept of compatibility refers to the capacity of new materials to interact with the substrate materials (original or substitution materials) without causing any damage. Different material characteristics, such as surface features (color, texture), chemical and

mineralogical composition, mechanical and physical properties, such as porosity, thermal dilation and hydric behavior, should be as similar as possible to the original substrate [9–12]. Other material characteristics, such as density, thermal expansion coefficient, elasticity modulus, and compressive strength have an essential role in the compatibility between mortars and substrates [13].

For Kozlowski et al. [14], the choice of restoration mortars in cultural heritage must be based on compatibility criteria, including functional and aesthetic aspects. The mortars should not accelerate the damage of old masonry by introducing stresses, retaining water, or favoring harmful chemical reactions. The most direct method to achieve compatibility is to use new mortars with compositions close to the old.

Many studies have investigated the degree of stone/mortar compatibility [9,11,14]. For Van Balen et al. [11], a lack of compatibility leads to damage caused by differences in physical, chemical, and mechanical properties that reduce the durability of the mortar/substrate composite. The requirements for compatible materials between a mortar and a substrate are surface features, composition, grain size distribution, compressive strength, elasticity, porosity properties, and thermal dilation coefficient.

Consequences of an inadequate selection of restoration mortars could lead to a loss of the original material or the degradation of the original substrate, masonry unit hardness issues, chemical processes altering material properties, and moisture transfer issues caused by incompatibility between the pre-existing elements and the new materials [14,15]. The determination of the compatibility of restoration mortars with substrates and adjacent mortars requires a complete set of testing procedures, such as mechanical testing, water intake, capillary action, setting/drying, and in situ adhesion testing [9]. Several authors have discussed the compatibility requirements for restoration mortars. Most of the researchers set requirements but do not discuss the acceptable level of differences between mortar and stone properties. Isebaert et al. [9] make a review of the compatibility requirements for restoration mortars and proposes a table with acceptance percentages.

The mechanical strength of the mortar should not be greater than that of the stone, and both values should be in the same order of magnitude [10]. A lower capillary absorption can decrease the entrance of moisture but also produce an accumulation of water in the interface during the drying process. The water presence and the channels through which it can pass are conversely beneficial for preservation due to the natural ability to heal fissures by lime carbonatation: water in cracks carries the free lime solution to the surface, where it meets air and hardens [15,16]. A higher value of the capillarity coefficient can cause a decay process. The natural capillarity of masonry typically allows moisture to enter but not fully penetrate walls. When attempting to escape the masonry, the water would commonly escape through the mortar joint, the stone, or the brick. A high capillarity causes moisture to pass in greater volume that can cause moisture retention, putting the masonry at a more susceptible state that encourages salt formations and problematic freeze/thaw issues [17]. Therefore, mortars/substrate compatibility criteria are always related to physical parameters, such as porosity and hydric properties. The absence of salts or any harmful chemical element in the mortar/substrate is essential to make a successful restoration.

For aesthetical reasons, perceptible color is an important characteristic in the restoration of historical monuments, achieving a restoration where the differences in color between the new materials and the old materials are not perceived is a difficult task. That is why one of the basic criteria in restoration is the color difference with the original material. The color differences can be evaluated using the total color variation ($\Delta E^*$), measured from color coordinates $L^*$ (lightness), $a^*$ (red/green coordinate), and $b^*$ (yellow/blue coordinate) [18]. If $\Delta E^*$ is lower than 1, it is not possible to notice a color difference; if it is greater than 3.5, it is possible to notice a difference; and if it between 5 and 15, the difference is evident.

The term durability refers to the ability of materials to resist weathering processes and their capacity to retain over time their shape, size, and aesthetic appearance, in addition to their physical and chemical properties, such as strength, mineralogy, or porosity [19] The principal agent of weathering is water coming by two main mechanisms: the capillarity

rises of underground water and rainwater infiltration. Their quantity and location in the building depend on the type of stone, the wall orientation, and the environmental conditions [20,21]. The most accepted ways of measuring durability in masonry materials are freezing thawing phenomena and soluble salts crystallization. There are many variants that interact in the durability of a material and especially in the composition of a mortar/substrate; Cultrone et al. carried out a study considering the drying kinetics of the materials, the size of the pores, the pH of the contact areas and the hydric properties [22].

There is a clear need to develop a methodology to select an adequate restoration mortar to be employed in any given substrate using objective and quantitative criteria. Several papers have proposed methodologies and criteria for the selection of restoration mortars according to specific locations or trying to reproduce ancient mortars. During the last 20 years, the holistic methodology developed in the Laboratory of Building Materials of the Aristotle University of Thessaloniki (Greece), has been used in more than 300 monuments with more than 2000 mortar samples. This methodology considers the environmental conditions and the analysis of the building, following the physical, chemical, and mechanical properties of original mortars [23]. Reverse engineering has been used to ensure compatibility and performance requirements using local materials and imitating ancient mortars. [13,24]. This procedure uses the determination of total soluble salts, X-ray diffraction, optical microscopy, SEM, DTA, FT-IR, and no destructive techniques to calculate the physical properties to compare ancient and new materials.

An important aspect that is not usually considered in the selection of restoration mortars is its impact on the environment. Life Cycle Assessment (LCA) is a standardized (ISO 14040 series) and well accepted methodology to estimate the total environmental impact of a system or product during its lifespan [25,26]. Among the many environmental impacts that can be quantified, the most known and measured is the Global Warming Potential (GWP), which is proportional to the amount of $CO_2$-equivalent ($CO_2$eq) emissions in the product life cycle. Lime binders are interesting from a carbon capture perspective [27] and their GWP have been assessed in several research works [28,29].

This paper aims to propose an adaptable, easy to use, universal methodology for the selection of restoration mortars. This methodology gives a first approach for the selection of the best mortar for a specific restoration work according to some mortars and substrate properties. The paper also presents the design and testing of sustainable formulations of lime restoration mortars. The innovative selection procedure is based on compatibility and sustainability criteria, including environmental impact, durability, and economic cost. The application to a case study, an historic monument in Paris built with lutetian and Euville limestone, is used to showcase the application of the method. For other restoration works, other mortars can be formulated and tested, according to the substrate, the traditional mortars employed in the building, the environment, the climatic conditions, the orientation of the wall, etc. The goal of this methodology is to be adaptable to different cases, with different supports (limestone, sandstone, bricks, et.), and located in different environments.

## 2. Materials and Methods

The study involved the design of 24 restoration mortars for historical monuments, adapted for small or semi-industrial manufactures. In these formulations, different aggregates, binders, admixtures, and additives were employed:

- Three binders: aerial lime CL90S (A), and two natural hydraulic limes NHL3.5 (H3.5) and NHL5 (H) from Socli Company part of Italcementi Group;
- Four types of aggregates with 0–2 mm: silica sand from Sacamat (S), calcareous sand from BPE Leciuex (C), silico-calcareous sand from Italcementi Group (D), and fine silica sand from Sibelco (F);
- Two recycled admixtures prepared in the laboratory by grinding in a tungsten miller: crushed brick (B) from Briqueterie d'Allone, and glass (G) from recycled glass provided by the Fédération du Verre;

- Two natural additives: Pine Cone from the woods of the Paris area, ground in a kitchen blender (P), and pine oil extracted in water solution resin using a kitchen steamer (R).

*2.1. Substrates*

2.1.1. Lutetian Limestone

The term 'Lutetian' was introduced by nineteenth-century geologists, derived from the Roman name for Paris: Lutetia, sometimes called Saint-Maximin or Saint-Leu stone, is a limestone from the Paris basin deposited in the warm sea that covered the Paris region approximately 45 million years ago (Eocen). Already very present in Gallo-Roman times, stone mining is one of the oldest industries in the South of the Oise [8,30]. Limestone with milioles and nummulites with an appearance of a plain beige background, fine, occasional medium grain, and large shells [30], one of the largest building stones of the centre of Paris [31].

ROCAMAT company donated the stone plates used, extracted in the quarries of Saint-Vaast les Mello (49°13′14.5″ N 2°26′59.8″ E), located approximately 60 km from Paris [30].

2.1.2. Euville Limestone

Euville limestone (EU) of age Oxfordian (Late Jurassic) is a building stone widely used in Belgium and France [32,33]. It is a white-beige to pink color grainstone composed of large crinoid fragments between 0.5 and 2 mm in length. The average diameter is approximately 900 μm. The formation around Euville is characterized by a complex succession of coral bioherms, separated by inter-reef zones. Fragments of Echinodermata (sea urchins), brachiopods, coral, and pellets can be found in this building limestone [34]. The fossils have syntaxial over-growth of calcite. It is almost completely (98%) composed of calcium carbonate with and appearance of a deep beige color, angular and sparkling medium grain [35].

Euville limestone quarries are located near Commercy (Département de la Meuse, France). It is mined in Euville, Géville and Commercy, Sorcy, Lérouville, and the Meuse coast, in the Meuse department. The samples used in this work are derived from the ROCAMAT (www.rocamat.com, accessed on 16 December 2021) in the quarries at 60 km of the Meuse river in the geolocation 47°43′41.4″ N 4°13′33.7″ E [36].

2.1.3. Mayan Stones

The territory of the Yucatan peninsula, from a geological point of view, is a calcareous platform emerged from the sea, the geological age of the stone substrate tends to increase toward the south [37].

The surface of the state of Yucatan is composed mainly of limestone from the Cenozoic era. The oldest rocks correspond to the Paleocene-Eocene epoch and are dolomitized, silicified, or recrystallized, made up of gypsum, anhydrite, halite, sulfates, and sodium chlorides [37–39].

The calcareous shell, locally known by the names of laja or chaltún (in Mayan) is extremely hard and constitutes the surface of the relief in large territories. There is also soft limestone that bears the Mayan name sahcab "white land". This soft material corresponds to unconsolidated rocks since the process of crystallization of aragonite to calcite did not occur. The materials that form the geological substratum in the Yucatan Peninsula are predominantly Tertiary and to a lesser extent Quaternary formation. There is a series of calcareous formations typical of the geology of the territory of the peninsula [37–39].

The stones are composed of calcium carbonate in more than 90% and with the appearance of deep beige and grey color.

*2.2. Mortar Formulation*

The 24 formulations are described in Table 1. Formulations 1 to 10, in green, correspond to mortars with natural hydraulic lime NHL5; formulations 11 to 16, in blue, correspond

to mortars with aerial lime; and formulations 17 to 24, in orange, correspond to mortars with natural hydraulic lime NHL3.5. The design of mortars was based on a trial and error procedure. Several properties were measured to characterize the mortars, such as compressive strength, porosity, density, capillarity, color, durability, etc.

**Table 1.** Restoration mortar formulations ($\pm$1% $w/w$). G: Ground waste glass, PN: Pinecone, R (pinecone resin solution indicated as *), B: Ground brick waste grounded, Sand: S (Sacamat), C (BPE Leciuex), D (Italcementi group) and F (Sibelco).

| # | Type of Mortar | Binders | | | Aggregates | | | | Additives | | | |
|---|---|---|---|---|---|---|---|---|---|---|---|---|
| | | NHL5 | NHL3.5 | CL90 | Sand D | Sand F | Sand S | Sand C | G | PN | R | B |
| 1 | HFD | 20 | - | - | 60 | 20 | - | - | - | - | - | - |
| 2 | HSD | 20 | - | - | 15 | - | 65 | - | - | - | - | - |
| 3 | HS | 20 | - | - | - | - | 80 | - | - | - | - | - |
| 4 | HB | 20 | - | - | - | - | - | - | - | - | - | 80 |
| 5 | HCSR | 30 | - | - | - | - | 35 | 35 | - | - | * | - |
| 6 | HCS | 30 | - | - | - | - | 35 | 35 | - | - | - | - |
| 7 | HSG | 30 | - | - | - | - | 60 | - | 10 | - | - | - |
| 8 | HSP | 30 | - | - | - | - | 68 | - | - | 2 | - | - |
| 9 | HC | 30 | - | - | - | - | - | 70 | - | - | - | - |
| 10 | HCSGB | 30 | - | - | - | - | 25 | 25 | 10 | - | - | 10 |
| 11 | AS | - | - | 20 | - | - | 80 | - | - | - | - | - |
| 12 | AHCS | 15 | - | 15 | - | - | 35 | 35 | - | - | - | - |
| 13 | ACSGB | - | - | 30 | - | - | 25 | 25 | 10 | - | - | 10 |
| 14 | OAC * | - | - | 40 | - | - | - | 60 | - | - | - | - |
| 15 | AC | - | - | 40 | - | - | - | 60 | - | - | - | - |
| 16 | AS2 | - | - | 40 | - | - | 60 | - | - | - | - | - |
| 17 | H3.5CS | - | 30 | - | - | - | 7 | 63 | - | - | - | - |
| 18 | H3.5CS2 | - | 30 | - | - | - | 21 | 49 | - | - | - | - |
| 19 | H3.5CS3 | - | 30 | - | - | - | 35 | 35 | - | - | - | - |
| 20 | H3.5CS4 | - | 30 | - | - | - | 49 | 21 | - | - | - | - |
| 21 | H3.5CS5 | - | 30 | - | - | - | 63 | 7 | - | - | - | - |
| 22 | H3.5CSG | - | 30 | - | - | - | 30 | 30 | 10 | - | - | - |
| 23 | H3.5CSB | - | 30 | - | - | - | 30 | 30 | - | - | - | 10 |
| 24 | H3.5CSGB | - | 30 | - | - | - | 25 | 25 | 10 | - | - | 10 |

The effects of granulometry were first tested to find an adequate aggregate's size distribution. Different sizes of sands were used, sand F being the finest, followed by sands S and C (0 to 2 cm) and sand D being the coarsest (0 to 4 cm diameter). The aggregate size was adapted by sieving, to obtain granulometric curves similar to those of commercial mortars. The first mortar was done with fine and coarse calcareous silica mix sand, and with 20% of binder NHL5 (HFD). The next formulated mortars were HSD, with the same binder but a mixture of medium and coarse sand, HC, and HS.

After testing the granulometry in different mixtures following the trial and error methodology, the next mortars were created with the S and C sands because they had a similar granulometry to commercial mortars. To test the effects that additives/admixtures can cause in mortars, we started with a binder, in this case, NHL5 and mixed it with different sand and additive/admixture; thus, HSG, HSP, and HCSR were formulated.

Mortars HCS and the series H3.5CS1-5 were planned to study the effect of a mix of sands on properties. In a similar way, AHCS was created to study the effect on properties of mortars of a mix of binders.

Considering that during restoration, subtract could be bricks or a mix of stone and brick, mortars with ground bricks as aggregate or as admixture were elaborated (HB, H3.5CSB). Mortars H3.5CSG and H3.5CSB were created to evaluate the properties caused by additives/admixtures on mortars properties.

In ancient mortars, it was a common practice to use organic additives. The mortar HCSR was created to study the effect caused by pinecones resin as an additive in the properties of the mortars.

The ancient techniques of lime preparation used to put the lime in water for a long time (lime putty) inspired the formulation of OAC. Other mortars were thought to compare the properties of lime putty mortar: AC mortar with powdered industrial lime and calcareous sand, and AS and AS2 mortars with powdered industrial lime and silica sand.

Finally, HCSGB, ACSGB, and H3.5CSGB were created to compare the effects of different binders on mortars with the same mix of sands and admixture.

Mortars were prepared in the laboratory at 24 °C and 40% relative humidity. Sands were dried at 60 °C for 24 h, and premixed with the binder for 1 min in an electric mixer (Rubimix 9, Rubi, Hialeah, FL, USA). Then, water was added to the powder and mixed for 5 min. The amount of water was varied to obtain the same consistency for all the mortars (Table 2). The consistency was measured with the reduced slump cone test [40–42], were the range of 1 to 4 cm corresponds to a soft-plastic consistency. Next, the mortar was molded in prismatic casts (40 mm × 40 mm × 160 mm) and then kept in plastic boxes for 7 days at a relative humidity of 90% (±5%). For air lime, the conditions were 5 days in the mold and 2 days without mold, for hydraulic lime it was 2 days with and 5 days without the mold, then storage in a humidity chamber at 65 ± 5% for 21 days according to EN 1015-2 standard [43]. Finally, they were stored under laboratory conditions. In order to evaluate the physical and mechanical properties in dry and carbonated samples, all the tests were carried out after 180 days at room conditions (25 °C and 45% relative humidity).

**Table 2.** Mixing water and consistency in restoration mortar formulations, *W/B*: water/binder ratio, i.e., water/lime ratio.

| # | Type of Mortar | Lime:Sand Ratio (By Weight) | Water (*W/B* Ratio) | Slump (cm) | Consistency Class |
|---|---|---|---|---|---|
| 1 | HFD | 1:4 | 0.80 | 1.4 | Soft-plastic |
| 2 | HSD | 1:4 | 0.90 | 1.6 | Soft-plastic |
| 3 | HS | 1:4 | 0.70 | 1.6 | Soft-plastic |
| 4 | HB | 1:4 | 1.35 | 1.5 | Soft-plastic |
| 5 | HCSR | 1:2.3 | 0.50 | 1.6 | Soft-plastic |
| 6 | HCS | 1:2.3 | 0.50 | 1.6 | Soft-plastic |
| 7 | HSG | 1:2.3 | 0.50 | 1.7 | Soft-plastic |
| 8 | HSP | 1:2.3 | 0.50 | 2.1 | Soft-plastic |
| 9 | HC | 1:2.3 | 0.47 | 1.8 | Soft-plastic |
| 10 | HCSGB | 1:2.3 | 0.69 | 2 | Soft-plastic |
| 11 | AS | 1:4 | 0.90 | 2 | Soft-plastic |
| 12 | AHCS | 1:2.3 | 0.73 | 1.2 | Soft-plastic |
| 13 | ACSGB | 1:2.3 | 1.06 | 2.1 | Soft-plastic |
| 14 | OAC * | 1:2.5 | 1.40 | 2.8 | Soft-plastic |
| 15 | AC | 1:2.5 | 0.94 | 2.4 | Soft-plastic |
| 16 | AS2 | 1:2.5 | 0.89 | 2.1 | Soft-plastic |
| 17 | H3.5CS | 1:2.3 | 0.87 | 1.3 | Soft-plastic |
| 18 | H3.5CS2 | 1:2.3 | 0.78 | 2.2 | Soft-plastic |
| 19 | H3.5CS3 | 1:2.3 | 0.77 | 1.4 | Soft-plastic |
| 20 | H3.5CS4 | 1:2.3 | 0.77 | 1.3 | Soft-plastic |
| 21 | H3.5CS5 | 1:2.3 | 0.76 | 1.9 | Soft-plastic |
| 22 | H3.5CSG | 1:2.3 | 0.69 | 2.3 | Soft-plastic |
| 23 | H3.5CSB | 1:2.3 | 0.73 | 1.4 | Soft-plastic |
| 24 | H3.5CSGB | 1:2.3 | 0.77 | 2.2 | Soft-plastic |

*2.3. Measured Properties*

2.3.1. Color

Color was measured using a Konica Minolta CM-2300d (Tokyo, Japan) spectrophotometer with a spot test area of 8 mm in the visible region (360–740 nm), spectral resolution of 10 nm in SCI mode with diffuse illumination, 8-degree viewing angle, using D65 as illuminant of reference and expressed in the CIE *L\*a\*b\** color system. Color difference (Δ*E\**),

between the substrate (subindex 1) and the restoration mortar (subindex 2) was measured according to:

$$\Delta E^* = \sqrt{\left(a_1^* - a_2^*\right)^2 + \left(b_1^* - b_2^*\right)^2 + \left(L_1^* - L_2^*\right)^2} \tag{1}$$

$\Delta L^*$ = difference in lightness between the sample and the standard color. (+ = lighter, − = darker). $\Delta a^*$ = difference in $a^*$ color coordinates between the sample and the standard, (+ = redder, − = greener). $\Delta b^*$ difference in $b^*$ color coordinates between the sample and the standard, (+ = yellower, − = bluer).

### 2.3.2. Porosity and Density

Porosity and density were measured by using the EN 1936 standard. Briefly, the samples were saturated under vacuum with deionized water, then the dry, saturated, and immersed mass were measured and the density and porosity were calculated according to:

$$\rho_b = \frac{Md}{Ms - Mh}\, \rho\ water \tag{2}$$

$$\varnothing = \frac{Ms - Md}{Ms - Mh}\, 100 \tag{3}$$

where $Md$ is the mass of the dry sample, $Ms$ is the mass of the sample measured after vacuum saturation with water, and $Mh$ is the mass of the saturated sample immersed in water; $\rho_b$ is the apparent density and $\varnothing$ is the % of open porosity.

### 2.3.3. Compressive and Flexural Strength

The compressive and flexural strength was obtained according to EN 1015-11 standard [44] using an INSTRON press with a maximum load of 20 kN. Compressive strength was tested in cubes of 4 cm using a GDS press with a maximum load of 100 kN.

### 2.3.4. Durability

The durability of the samples was estimated by salt crystallization and freezing thawing cycles, adapted from standards EN 12371 and EN 12370 [45,46]. All tests were made in triplicate. In the durability to freezing thawing tests, samples were dried for 24 h at 60 °C, weighed and then immersed in water at a constant temperature of $20 \pm 0.5$ °C for 2 h. The samples were removed from the water and placed in a freezer at −15 °C in a dry condition for 15 h. The samples were subjected to 20 cycles. On the other hand, during the salt crystallization test, samples were dried for 24 h at 60 °C, weighed and then immersed in a 14% ($w/w$) $Na_2SO_4$ 10 $H_2O$ solution at a constant temperature of $20 \pm 0.5$ °C for 2 h. The samples were removed from the solution and dried in a pre-heated oven at 60 °C for 15 h. The samples were subjected to 16 cycles.

### 2.3.5. Environmental Impact: Global Warming Potential

The GWP to 100 years was calculated using the CML-IA method [47]. The life cycle inventory was modeled in the OpenLCA software, using primary data for direct inputs and the European Reference Life Cycle Database (ELCD) V3.2 for background processes. The complete inventory data and LCA assumptions are discussed in previous works on restoration mortars [48,49]. Results of these detailed LCAs were fitted to Equation (4), by using only the dominant contributions to GWP, as a means of having a tool to quickly estimate the GWP of restoration mortars.

The emission factors (*EFs*) for binders, aggregates, and other essential inputs, required to use Equation (4) are listed in Table 3. These *EFs* include those of silica sand production [50], lime production [51–53], and calcareous sand production [54,55]. An example of the calculation is presented in the Supplementary Materials in Table S5.

$$GWP\left(\mathrm{kg\,CO_{2\ eq}/t}\right) = \sum M_i EF_i \tag{4}$$

**Table 3.** Emission factors of inputs [42].

| Inputs | Unit | *EF* |
|---|---|---|
| Portland Cement | kg $CO_2$ eq/kg | 1.53 |
| CL90 | kg $CO_2$ eq/kg | 0.98 |
| NHL5 | kg $CO_2$ eq/kg | 0.74 |
| NHL3.5 | kg $CO_2$ eq/kg | 0.64 |
| Silica Sand | kg $CO_2$ eq/kg | 0.045 |
| Calcareous Sand | kg $CO_2$ eq/kg | 0.03 |
| Silico-Calcareous Sand | kg $CO_2$ eq/kg | 0.05 |
| Fine Silica Sand | kg $CO_2$ eq/kg | 0.07 |
| Transport (raw materials) | kg $CO_2$ eq/tkm | 0.368 |
| Transport (disposal) | kg $CO_2$ eq/tkm | 0.368 |

In Equation (4), $M_i$ are the amounts of input *i* (mass or energy) in the mortar formulation, and $EF_i$ are the emission factors from Table 3. In the case of transport inputs, the flow is usually expressed in units of work (tkm). In these calculations, special attention was given to transport processes, for both raw materials (from the manufacture place to the restoration site) and for disposal at the end of life (from the restoration site to the final disposal).

*2.4. Methodology for the Calculation of Scores*

Due to the wide variety of building materials and techniques used in different epochs and geographic zones, a versatile material to restore all types of existing buildings does not exist. The designed selection procedure (Figure 1) can be used anywhere to help restorers to select or design the best restoration mortars specially formulated for the intervention in the stone wall, ornaments, or sculptures, according to chemical, physical, aesthetical, and environmental requirements.

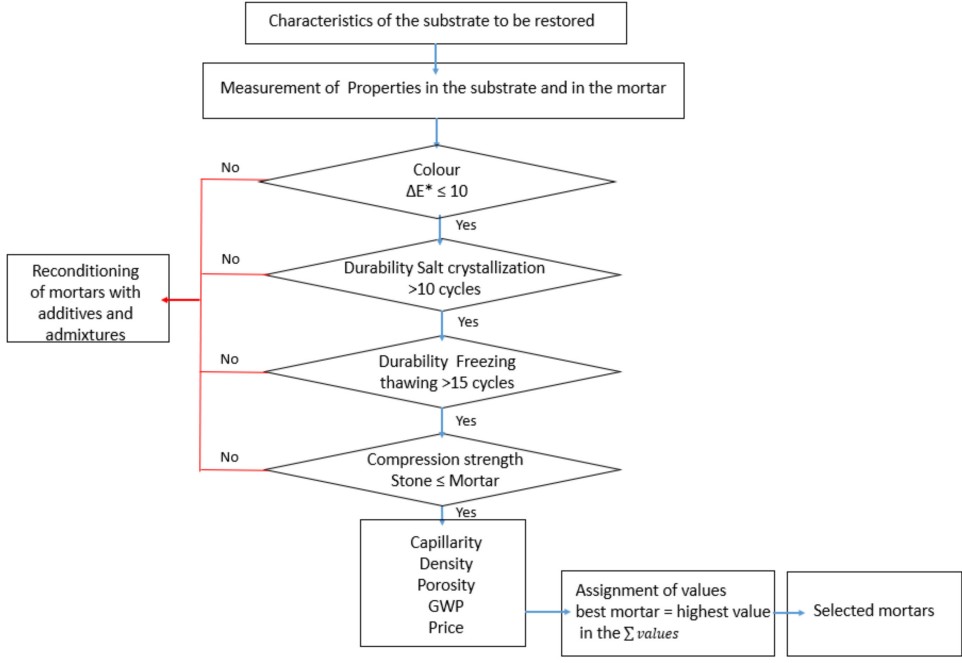

**Figure 1.** Diagram of the proposed methodology for the selection of sustainable restoration mortars.

This methodology sets a score for any mortar considering chemical, physical, aesthetical, and ecological requirements. A normalization equation for each property is proposed to assign a value between −100 and 100. In this way, the original value of the property is normalized in an "adequacy," dimensionless scale that allows adding together the values to obtain a final score. The proposed equation includes weighing factors to each property

that might be adapted depending on the user's expert judgment. This structure also allows for the inclusion of additional properties, if the user deems it necessary for particular applications. At the end of the calculations, the mortars with the highest scores are the best rated options.

The proposed properties to assess restoration mortars are:

Property 1. The color difference $\Delta E^*$ between mortar and substrate must be the lowest possible. If $\Delta E^* > 10$, the color difference will be too evident and thus the mortar needs to be reformulated before giving it a score. It is recommended to use pigments to get $\Delta E^* < 3$. For the value assignment, if the $\Delta E^*$ is equal or higher than 10, this property gets 0 points. Equation (5) considers an indicator range of $0 \le Co \le 100$.

$$Co = \frac{10 - \Delta E^*}{0.1} \tag{5}$$

Property 2. Compressive strength (in MPa) of the mortar must be lower than that of the substrate. If this condition is not fulfilled, the mortar cannot be used and a reformulation is required; the amount of binder may be reduced to lower the compressive strength. If the mortar's compressive strength is 10 MPa lower than the stone's but strong enough to be a restoration mortar, this property gets 0 points. Equation (6) considers an indicator range of $0 \le CS \le 100$ as follows:

$$CS = \left( \frac{10 - (Ss - Sm)}{0.1} \right) \tag{6}$$

where $Ss$ is the compression strength of the substrate and $Sm$ is the compression strength of the mortar.

Property 3. The mortars must be as durable as possible. In case that the durability of the mortar is very low (loss of more than 80% of weight in less than 10 cycles), additives can be used to improve it (Table S7). The value is assigned using the mass difference in the last cycle of the crystallization test ($\Delta M$), and $M$ is the original mass. Equation (7) considers an indicator range of $0 \le D1 \le 100$.

$$D1 = \left( 1 - \left| \frac{\Delta M}{M} \right| \right) \times 100 \tag{7}$$

Property 4. The mortars must be as durable as possible in the freezing thawing cycles test [38]. Additives can be used to improve their durability (Table S7). The value is assigned by the mass difference in the last cycle. Equation (8) considers an indicator range $0 \le D2 \le 100$.

$$D2 = \left( 1 - \left| \frac{\Delta M}{M} \right| \right) \times 100 \tag{8}$$

where $\Delta M/M$ have the same definitions as Equation (5).

Property 5. The capillarity coefficient ($kg/m^2s^{1/2}$) of the mortar must be as close as possible to that of the substrate. Equation (9) has an indicator range of $0 \le CA \le 100$.

$$CA = (1 - |Cs - Cm|) \times 100 \tag{9}$$

where Cm is the capillarity coefficient of the mortar and Cs is the capillarity coefficient of the substrate.

Property 6. Density (in $kg/m^3$) of the mortar ($\rho m$) and the substrate ($\rho s$) must be as similar as possible. Equation (10) was used for value assignment of the density difference which has an indicator range of $0 \le DE \le 100$.

$$DE = \frac{|1000 - |\rho s - \rho m||}{10} \tag{10}$$

Property 7. The porosity (in percentage) of the mortar (øm) must be as close as possible to that of the substrate (øs). Equation (11) considers an indicator range of $0 \leq PO \leq 100$.

$$PO = (1 - |øs - øm|) \times 100 \tag{11}$$

Property 8. The environmental performance of mortars is evaluated with the GWP. It has been shown that other environmental impacts in the life cycle of mortars correlate well with the GWP, including energy consumption, acidification, photochemical oxidation, and toxicity potential [49]. Hence, GWP is a good indicator of the overall life cycle environmental impacts. The simplified GWP of mortar production is calculated using Equation (2). As a reference, the GWP of air lime is approximately 1000 kg $CO_2$eq/t and all mortars are expected to have a lower value than that. Thus, the value assignment in Equation (12) considers an indicator range of $0 \leq GW \leq 100$:

$$GW = \frac{1000 - GWP}{1000}(100) \tag{12}$$

Property 9. The cost of raw materials is a good proxy for the cost of the production per ton of mortar on site. A maximum value of 800 €/t was considered (price of a ton of air lime) was considered in Equation (13), which has an indicator range of $0 \leq P \leq 100$:

$$P = \frac{800 - €/t}{800}(100) \tag{13}$$

The final score of the evaluated mortars will be the sum of the dimensionless values calculated in Equations (5) to (13), as follows:

$$Score = Co + CA + CS + D1 + D2 + DE + PO + GW + P \tag{14}$$

Equation (12), can be modified if one property is considered to be more important than another. In our case, we consider that the coefficient of capillarity and the durability are essential properties to ensure cohesion between mortar and the substrate [56,57] and to be preserved in time. In addition, GWP is especially important as many environmental impacts are represented within this indicator. According to these considerations, we propose to include in Equation (14) a weighing factor of 2 for these three variables, also follows:

$$Score\ (weighed) = Co + 2CA + CS + 2D1 + 2D2 + DE + PO + 2GW + P \tag{15}$$

In order to illustrate the application of this selection method, we simulate a restoration exercise of the Royal Palace (Paris) built with Lutetian stone (Figure 2). Experimental mortars were compared with the commercial (Altar Pierre, Artopierre and Lithomex), previously studied in our laboratory and largely employed in France and other parts of the world for restoration works [12].

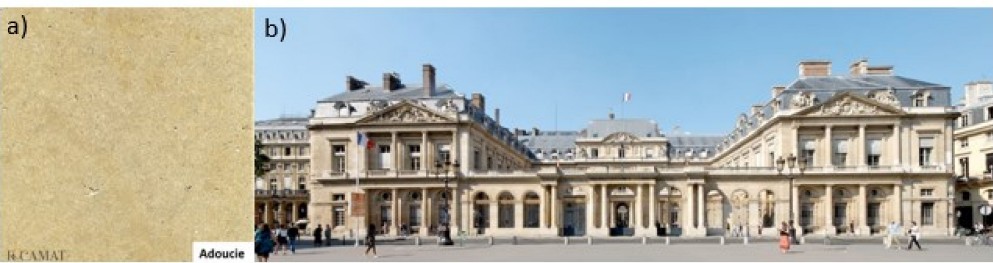

**Figure 2.** (**a**) Lutetian stone, (**b**) The Royal Palace of Paris, France.

A sensitivity analysis determines the extension of the inputs uncertainty in a mathematical model [58]. It is a way to predict the outcome of a decision given by a specific range of variables [43]. The scenarios of the sensitivity analysis are presented in Table 4.

The low, average and high values in the score were defined according to literature and for HC mortar elaborated with 2 ingredients: 30% of natural hydraulic lime 5 and 70% of calcareous sand whose values were taken as base values for the study.

**Table 4.** Scenarios for the sensitivity analysis.

| Variable (Units) | Low | Base Value (HC) | High | References |
|---|---|---|---|---|
| | Value | | Value | |
| Color ($\Delta E^*$ colorimetric units) | 1 | 9.7 | 15 | [59] |
| Capillarity ($kg/m^2s^{1/2}$) | 0.17 | 0.3 | 0.37 | [57,60,61] |
| Compressive Strength (MP) | 2.2 | 4.7 | 7.8 | [9,13,60] |
| Durability (g) | −0.25 | −0.05 | 0 | [9,13] |
| Density ($kg/m^3$) | 1617 | 1673 | 1900 | [9,13] |
| Porosity (%) | 0.2 | 0.36 | 0.5 | [9,13,60] |
| GWP ($kg\ CO_2eq/t$) | 250 | 370 | 600 | [48] |
| Cost (€/t) | 150 | 187 | 250 | [62,63] |

## 3. Results and Discussion

Considering the color $\Delta E^*$ (color difference) with the Lutetian stone (Table 5), seven mortar formulations were selected as the first filter (highlighted in green). It should be considered that colorants have not been employed in the formulations used in this work. In a real case, if the aesthetic issue is important, pigments should be applied until $\Delta E^*$ has a value less than 3. Three commercial mortars were selected for comparison and included in Table 5. Other cases studies, such as Euville and Mayan stones, are available in Table S1.

**Table 5.** Color of restoration mortars for Lutetian stone and selected commercial mortars. $L^*$ = difference in lightness $a^*$ = difference redder to greener. $b^*$ = difference yellower to bluer.

| | Sample | $L^*$ | $a^*$ | $b^*$ | $\Delta E$ Lutetian |
|---|---|---|---|---|---|
| 1 | HFD | 74.52 | 2.63 | 10.04 | 9.2 |
| 2 | HSD | 74.14 | 2.72 | 11.52 | 8.1 |
| 3 | HS | 72.88 | 2.55 | 9.93 | 10.2 |
| 4 | HB | 62.85 | 10.71 | 11.47 | 18.8 |
| 5 | HCSR | 84.17 | 1.41 | 5.95 | 13.4 |
| 6 | HCS | 81.1 | 1.83 | 7.954 | 10.5 |
| 7 | HSG | 82.19 | 1.44 | 6.19 | 12.6 |
| 8 | HSP | 82.66 | 1.64 | 8.46 | 10.5 |
| 9 | HC | 82.19 | 2.12 | 9.11 | 9.7 |
| 10 | HCSGB | 81.74 | 3.19 | 9.13 | 9.4 |
| 11 | AS | 87.55 | 1.18 | 7.55 | 13.9 |
| 12 | AHCS | 83.09 | 1.69 | 8.7 | 10.5 |
| 13 | ACSGB | 87.05 | 2.38 | 5.93 | 14.7 |
| 14 | OAC * | 84.49 | 1.74 | 8.55 | 11.2 |
| 15 | AC | 86.53 | 1.81 | 8.35 | 12.5 |
| 16 | AS2 | 81.32 | 1.16 | 7.26 | 11.4 |
| 17 | H3.5CS | 84.21 | 2.22 | 9.79 | 10.0 |
| 18 | H3.5CS2 | 86.84 | 1.78 | 8.84 | 12.4 |
| 19 | H3.5CS3 | 87.94 | 1.25 | 7.42 | 14.2 |
| 20 | H3.5CS4 | 85.96 | 1.73 | 7.89 | 12.6 |
| 21 | H3.5CS5 | 87.65 | 1.31 | 7.22 | 14.2 |
| 22 | H3.5CSG | 84.94 | 1.82 | 10.04 | 10.3 |
| 23 | H3.5CSB | 83.28 | 3.43 | 9.76 | 9.5 |
| 24 | H3.5CSGB | 79.02 | 4.72 | 12.68 | 5.6 |
| 25 | Lithomex | 79.5 | 2.31 | 11.23 | 7.0 |
| 26 | Artropierre | 88.04 | 0.81 | 6.37 | 15.2 |
| 27 | Altar Pierre | 75.64 | 1.94 | 10.39 | 8.5 |
| 28 | Lutetian stone | 78.84 | 3.43 | 18.12 | - |

Concerning the mortars' durability in the salt crystallization test (Figure 3), formulation 24 (H3.5CSGB) suffered the lowest mass loss, followed by formulation 17 (HCSGB) and formulation 9 (HC). Results for Lutetian stone (red line) are shown for comparison. On the other hand, the worst behavior during this durability test corresponded to formulations 1 and 17 (HFD and H3.5CS3). Mortars that present a lower durability than stone could be considered as sacrificial mortars. The values for all the mortars can be observed in Figure S1.

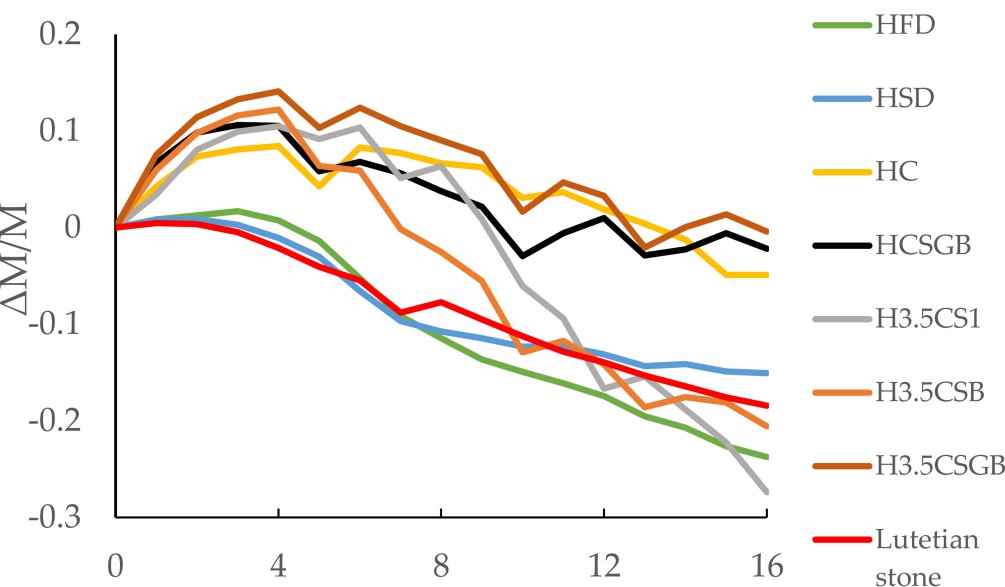

**Figure 3.** Result of the salt crystallization test for selected mortars.

Figure 4 shows the results of the durability tests in freezing thawing cycles. A similar behavior is observed, where HFD showed the greatest deterioration with approximately 10% of its mass. The results are homogeneous for mortars HCSGB, H3.5CS, H3.5CSB, and H3.5CSGB, but mortars HFD, HSD, and HC presented variations between the different cycles. The corresponding results of all other mortars are in Figure S2.

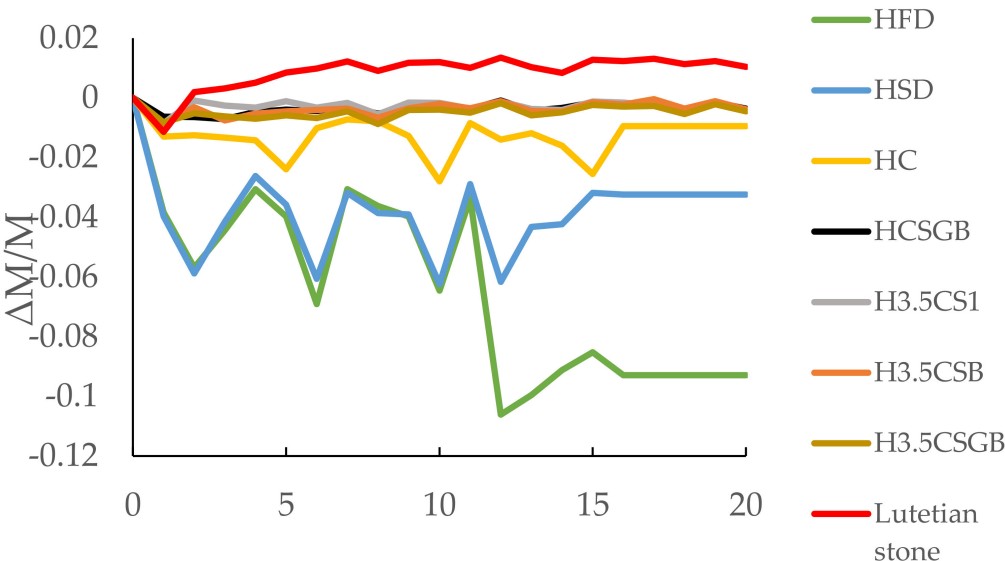

**Figure 4.** Experimental results from the freeze and thaw cycles test.

Following the selection methodology (Figure 1), the measured mechanical and physical properties are presented in Table 6.

**Table 6.** Physical and mechanical properties of selected mortars for Lutetian stone. $\rho b$: density (kg/m$^3$), $\varnothing$: open porosity (%), C: capillarity coefficient (kg/m$^2$s$^{1/2}$), CS 180: compression strength at 180 days (MPa), SD standard deviation.

|  | Mortar | $\rho$ | SD $\rho$ | $\varnothing$ | SD ø | C | SD C | S 180 | SD S-180 |
|---|---|---|---|---|---|---|---|---|---|
| 1 | HFD | 1873 | 5 | 30 | 2.8 | 0.32 | 0.03 | 2.3 | 0.12 |
| 2 | HSD | 1801 | 53 | 30 | 0.7 | 0.29 | 0.05 | 2.2 | 0.18 |
| 3 | HS | 1869 | 26 | 30 | 0.5 | 0.32 | 0.03 | 1.7 | 0.15 |
| 4 | HB | 1616 | 5 | 40 | 0.3 | 0.48 | 0.04 | 2.1 | 0.28 |
| 5 | HSCR | 1833 | 21 | 34 | 0.4 | 0.35 | 0.06 | 4.1 | 0.07 |
| 6 | HSC | 1846 | 4 | 33 | 0.7 | 0.21 | 0.02 | 3.7 | 0.97 |
| 7 | HSG | 1947 | 9 | 29 | 0.4 | 0.23 | 0.03 | 8.3 | 0.48 |
| 8 | HSP | 1673 | 9 | 38 | 7 | 0.22 | 0.08 | 0.7 | 0.19 |
| 9 | HC | 1772 | 96 | 36 | 0.3 | 0.3 | 0.03 | 4.7 | 1.03 |
| 10 | HCSGB | 1695 | 9 | 34 | 0.3 | 0.19 | 0.04 | 7.5 | 0.5 |
| 11 | AS | 1829 | 4 | 33 | 0.3 | 0.39 | 0.03 | 1.4 | 0.75 |
| 12 | AHCS | 1786 | 2 | 36 | 0.2 | 0.44 | 0.03 | 2.03 | 0.46 |
| 13 | ACSGB | 1572 | 20 | 37 | 0.8 | 0.48 | 0.04 | 1.5 | 0.45 |
| 14 | OAC * | 1653 | 4 | 40 | 0.2 | 0.49 | 0.04 | 1.6 | 0.16 |
| 15 | AC | 1506 | 1 | 45 | 1.1 | 0.58 | 0.05 | 2 | 0.26 |
| 16 | AS2 | 1586 | 37 | 42 | 1.1 | 0.53 | 0.03 | 1 | 0.12 |
| 17 | H3.5CS | 1688 | 24 | 40 | 0.9 | 0.53 | 0.06 | 2.55 | 0.19 |
| 18 | H3.5CS2 | 1668 | 13 | 41 | 1.2 | 0.53 | 0.04 | 2.19 | 0.51 |
| 19 | H3.5CS3 | 1743 | 2 | 37 | 0.2 | 0.43 | 0.06 | 3.04 | 0.15 |
| 20 | H3.5CS4 | 1741 | 5 | 37 | 0.9 | 0.5 | 0.02 | 1.89 | 0.17 |
| 21 | H3.5CS5 | 1780 | 6 | 36 | 0.4 | 0.47 | 0.03 | 2.15 | 0.32 |
| 22 | H3.5CSG | 1675 | 41 | 34 | 1.6 | 0.38 | 0.02 | 3.4 | 0.12 |
| 23 | H3.5CSB | 1689 | 32 | 35 | 1.3 | 0.33 | 0.04 | 3.5 | 0.98 |
| 24 | H3.5CSGB | 1647 | 39 | 37 | 0.5 | 0.39 | 0.03 | 4.8 | 0.35 |
| 25 | Lithomex | 1670 | 20 | 32 | 0.9 | 0.11 | 0.03 | 7 | 1.5 |
| 26 | Artropierre | 1574 | 15 | 39 | 0.8 | 0.47 | 0.03 | 7 | 1.35 |
| 27 | Altar Pierre | 1880 | 62 | 25 | 0.8 | 0.3 | 0.03 | 15.5 | 3.8 |
| 28 | Lutetian stone | 1617 | 19 | 37 | 0.8 | 0.35 | 0.04 | 8.5 | 0.83 |
| 29 | Euville stone | 1943 | 48 | 19 | 1.4 | 0.46 | 0.02 | 8.6 | 1.2 |

Regarding compression strength, all the mortars have lower values than the Lutetian stone. A lower value is recommended because larger values could damage the substrate.

Measured densities were between 1506 and 1947 kg/m$^3$. Specific samples had different values due to the properties of the additives or admixtures. It was observed that brick dust as admixture (B) decreases the density of mortars. It can be a consequence of a higher water absorption induced by the presence of clays or other compounds, such as albite and anorthite (Supplementary Material, Table S7). This behavior is illustrated in the mortars HB and H3.5CSB.

On the other hand, glass dust (G) used as admixture made the mortar more compact, raising its density as noticed in mortars HSG and H3.5CSG. No significant change was observed regarding the use of pinecone resin aqueous extracts, between HCSR and HCS mortar. On the other hand, the formulation with a 50/50 portion of calcareous sand/silica sand (H3.5CS3) has a higher compressive strength than the other mixtures.

The distance between the raw materials sources to the Royal Palace is shown in Table 7. These values were used to estimate the GWP with Equation (10).

**Table 7.** Transport distance of raw materials to the Royal Palace located at 8, Rue de Montpensier, 75001 Paris.

| Raw Material | Company | Factory Location | Distance (km) |
|---|---|---|---|
| Air Lime | Heidelberg Cement Group | Sauvanterre-la-lémance, 47500 | 590 |
| Hydraulic Lime | SOCLI Heidelberg Cement Group | Le Castans, 65370 | 790 |
| Calcareous Sand | BPE Leciuex | Saint Maximin, 60740 | 65 |
| Silica Sand | Sacamant France ZI | La forainte de Lannoy, 80120 | 205 |
| Silica-Calcareous Sand | GSM Heidelberg Cement Group | Guerville, 78931 | 55 |
| Silica-Fine Sand | Sibelco | Bourron-Marlotte, 77780 | 90 |

Figure 5 shows the results of the GWP. Mortar 4 (HB) has the lowest value (240 kg $CO_2$eq/t); however, this formulation was not considered due to the color difference. Among the selected mortars, formulation 1 (HFD) had the lowest GWP (294 kg $CO_2$eq/t), while formulation 10 (HCSGB) presented the highest (377 kg $CO_2$eq/t).

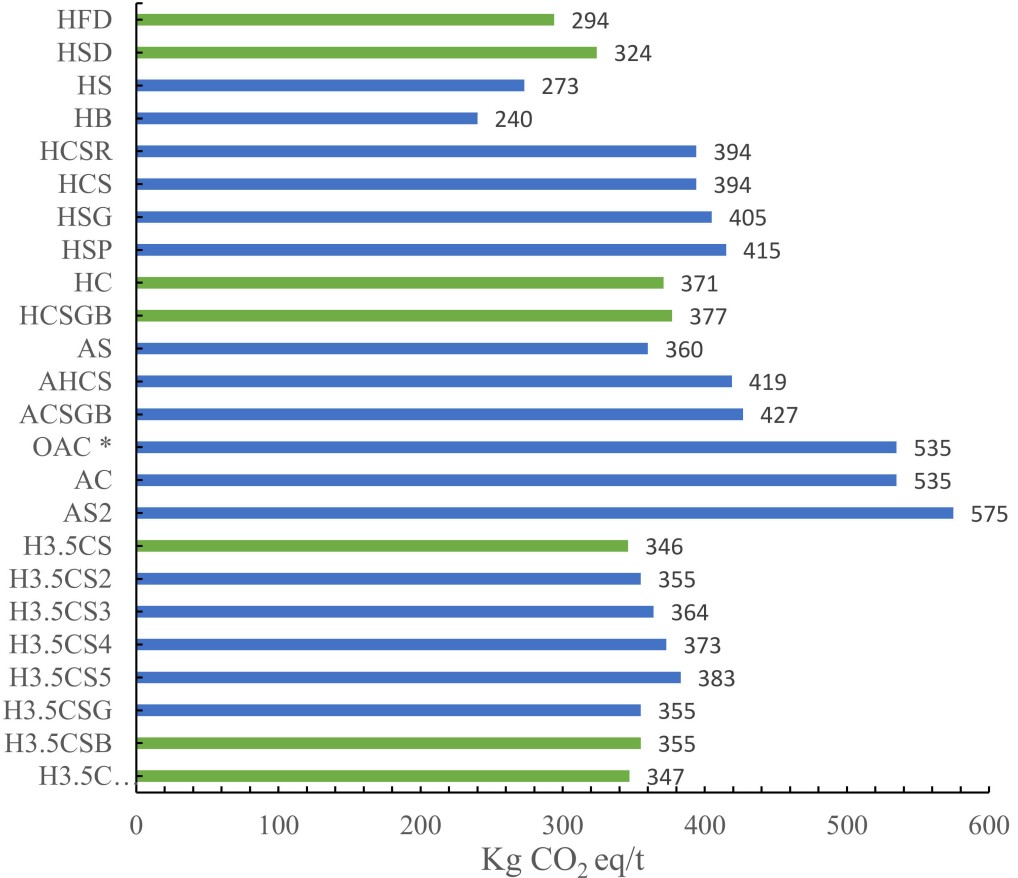

**Figure 5.** GWP of selected mortars.

Tables 8 and 9 present the costs of raw material acquisition per ton of mortar. This value does not include transport or processing, but for the sake of comparison between mortars, it is a good enough estimator.

**Table 8.** Cost (€/kg) of raw materials.

| | NHl5 | NHl3.5 | CL 90 | Sand F | Sand S | Sand C | Sand D | G | B |
|---|---|---|---|---|---|---|---|---|---|
| Price | 0.53 | 0.59 | 0.8 | 0.07 | 0.06 | 0.04 | 0.05 | 0 | 0 |

**Table 9.** Cost per ton (€/t) of mortars.

|  | Mortar Formulations | Price €/Ton |
| --- | --- | --- |
| 1 | HFD | 148 |
| 2 | HSD | 154 |
| 3 | HS | 154 |
| 4 | HB | 106 |
| 5 | HCSR | 194 |
| 6 | HCS | 194 |
| 7 | HSG | 195 |
| 8 | HSP | 200 |
| 9 | HC | 187 |
| 10 | HCSGB | 184 |
| 11 | AS | 208 |
| 12 | AHCS | 235 |
| 13 | ACSGB | 275 |
| 14 | OAC * | 344 |
| 15 | AC | 344 |
| 16 | AS2 | 356 |
| 17 | H3.5CS | 188 |
| 18 | H3.5CS2 | 209 |
| 19 | H3.5CS3 | 212 |
| 20 | H3.5CS4 | 215 |
| 21 | H3.5CS5 | 218 |
| 22 | H3.5CSG | 207 |
| 23 | H3.5CSB | 207 |
| 24 | H3.5CSGB | 202 |

The price of the commercial restoration mortars varies according to the distributor or the manufacturer. The average price in France in December 2021 is 3.4 €/kg for Lithomex [64], 1.94 €/kg for Artopierre [65] and 5.5 €/kg for Altar Pierre. Therefore, the price per ton is above 1900 €. As the exact formulation of commercial mortars is unknown, gross approximations were made to their GWP and production cost, for comparison purposes. Hence, the following values were assumed: GWP of 500 kg $CO_2$eq/t, cost of €500/t, and average durability (loss equal to or less than 15 percent in weight).

The values calculated by Equations (5) to (13) are presented in Table 10. The selected mortars appear in green and red for the mortars eliminated for having a compressive strength higher than the substrate's. As can be seen, the selected mortars present the highest values.

**Table 10.** Assignment of values for restoration mortars for Lutetian limestone.

| # | Mortar | Co | CA | CS | D1 | D2 | DE | PO | GW | P | Score |
| --- | --- | --- | --- | --- | --- | --- | --- | --- | --- | --- | --- |
| 1 | HFD | 8 | 97 | 38 | 76 | 98 | 74 | 93 | 71 | 82 | 637 |
| 2 | HSD | 19 | 94 | 37 | 85 | 92 | 81 | 93 | 68 | 81 | 649 |
| 3 | HS | 0 | 97 | 32 | 89 | 92 | 75 | 93 | 73 | 81 | 631 |
| 4 | HB | 0 | 87 | 36 | 96 | 85 | 100 | 97 | 76 | 87 | 664 |
| 5 | HCSR | 0 | 100 | 56 | 81 | 86 | 78 | 97 | 61 | 76 | 635 |
| 6 | HCS | 0 | 86 | 52 | 67 | 86 | 77 | 96 | 61 | 76 | 600 |
| 7 | HSG | 0 | 88 | 98 | 86 | 89 | 67 | 92 | 60 | 76 | 655 |
| 8 | HSP | 0 | 87 | 22 | 60 | 85 | 94 | 99 | 59 | 75 | 581 |
| 9 | HC | 3 | 95 | 62 | 95 | 87 | 85 | 99 | 63 | 77 | 665 |
| 10 | HCSGB | 6 | 84 | 90 | 98 | 91 | 92 | 97 | 62 | 77 | 698 |
| 11 | AS | 81 | 96 | 29 | 33 | 91 | 79 | 96 | 64 | 74 | 643 |
| 12 | AHCS | 0 | 91 | 35 | 44 | 86 | 83 | 99 | 58 | 71 | 567 |
| 13 | ACSGB | 0 | 87 | 30 | 71 | 82 | 96 | 100 | 57 | 66 | 588 |
| 14 | OAC * | 0 | 86 | 31 | 48 | 82 | 96 | 97 | 47 | 57 | 544 |
| 15 | AC | 0 | 77 | 35 | 25 | 87 | 89 | 92 | 47 | 57 | 508 |

**Table 10.** *Cont.*

| # | Mortar | Co | CA | CS | D1 | D2 | DE | PO | GW | P | Score |
|---|--------|----|----|----|----|----|----|----|----|---|-------|
| 16 | AS2 | 0 | 82 | 25 | 69 | 54 | 97 | 95 | 43 | 56 | 520 |
| 17 | H3.5CS | 0 | 82 | 41 | 41 | 82 | 93 | 97 | 65 | 77 | 577 |
| 18 | H3.5CS2 | 0 | 82 | 37 | 64 | 81 | 95 | 96 | 65 | 77 | 596 |
| 19 | H3.5CS3 | 0 | 92 | 45 | 73 | 84 | 87 | 100 | 64 | 74 | 619 |
| 20 | H3.5CS4 | 0 | 85 | 34 | 54 | 84 | 88 | 100 | 63 | 74 | 581 |
| 21 | H3.5CS5 | 0 | 88 | 37 | 76 | 85 | 84 | 99 | 62 | 73 | 603 |
| 22 | H3.5CSG | 0 | 97 | 49 | 84 | 84 | 94 | 97 | 65 | 73 | 642 |
| 23 | H3.5CSB | 5 | 98 | 50 | 79 | 84 | 93 | 98 | 65 | 74 | 645 |
| 24 | H3.5CSGl | 44 | 96 | 63 | 100 | 84 | 97 | 100 | 65 | 75 | 724 |
| 25 | Lithomex | 30 | 76 | 85 | 85 | 85 | 95 | 95 | 50 | 38 | 638 |
| 26 | Artropierre | 0 | 88 | 85 | 85 | 85 | 96 | 98 | 50 | 38 | 624 |
| 27 | Altapierre | 15 | 95 | 0 | 85 | 85 | 74 | 88 | 50 | 38 | 529 |

The selection procedure allows determining which mortars would be the most ecological, durable, and sustainable for the restoration of the Lutetian stone at the Royal Palace of Paris. They were 24 (H3.5CSGB), 10 (HCSGB), 2 (HSD) and 9 (HC).

To compare commercial and formulated mortars, Equation (15) was applied to the best options. Here, the coefficients were modified according to the importance of each property included in the Table 11. This is not an argument that discredits commercial mortars, it simply shows that mortars made specifically for a location in most cases will have better results. It should be noted that these formulated restoration mortars do not have additives, and some of their properties can still be improved by the use of specific additives.

**Table 11.** Score of selected mortars including weighing coefficients (Equation (15)).

| # | Mortar | Co | CA | CS | D1 | D2 | DE | PO | GW | P | Total |
|---|--------|----|----|----|----|----|----|----|----|---|-------|
| 1 | HFD | 8 | 97 | 38 | 76 | 98 | 74 | 93 | 71 | 82 | 978 |
| 2 | HSD | 19 | 94 | 37 | 85 | 92 | 81 | 93 | 68 | 81 | 988 |
| 3 | HS | −2 | 97 | 32 | 89 | 92 | 75 | 93 | 73 | 81 | 980 |
| 9 | HC | 3 | 95 | 62 | 95 | 87 | 85 | 99 | 63 | 77 | 1005 |
| 10 | HCSGB | 6 | 84 | 90 | 98 | 91 | 92 | 97 | 62 | 77 | 1033 |
| 22 | H3.5CSG | −3 | 97 | 90 | 84 | 84 | 94 | 97 | 65 | 73 | 1010 |
| 23 | H3.5CSB | 5 | 98 | 90 | 79 | 84 | 93 | 98 | 65 | 74 | 1011 |
| 24 | H3.5CSGB | 44 | 96 | 93 | 100 | 84 | 97 | 100 | 65 | 75 | 1099 |
| 25 | Lithomex | 30 | 76 | 85 | 90 | 90 | 95 | 95 | 62 | 63 | 1003 |
| 26 | Artropierre | 0 | 88 | 85 | 90 | 90 | 96 | 98 | 62 | 63 | 1001 |

The three best options were the mortar 24 (H3.5CSGB), 10 (HCSGB), and 23 (H3.5CSB).

Figure 6 presents the sensitivity study for the weighed score (Equation (15)). Results indicate that the largest sources of uncertainty are the color, cost, durability, and the GWP.

Color variations lead to the use of pigments that increase the GWP. The closer the color to the substrate, the more ecological the mortar. The color variation tends to increase the total value between −7% and 10%. In addition, if the cost is higher than 500 € per ton of mortar, the final variation can go up to −5%. The differences in durability can go from −5% to 1%, and the differences in GWP from −4% to 2%. Their influence in the final score is increased due to the weighing coefficient in Equation (15). However, the percentage they represent of the total makes it evident that they were considered but that their influence alone is less than 5 percent of the value of the total score. In conclusion, the presented methodology is fair, and the properties are well represented.

The most important difference between formulated mortars and commercial mortars is observed in their compressive strength, generally lower in formulated mortars than in the commercial, except for formulations 14 (8.6 MPa) and 17 (7.5 Mpa). The latter present values in the same range as those of commercial restoration mortars: Lithomex (7 to 9 Mpa), Artopierre (8 Mpa), and Altar Pierre (15.6 MPa) [12,66].

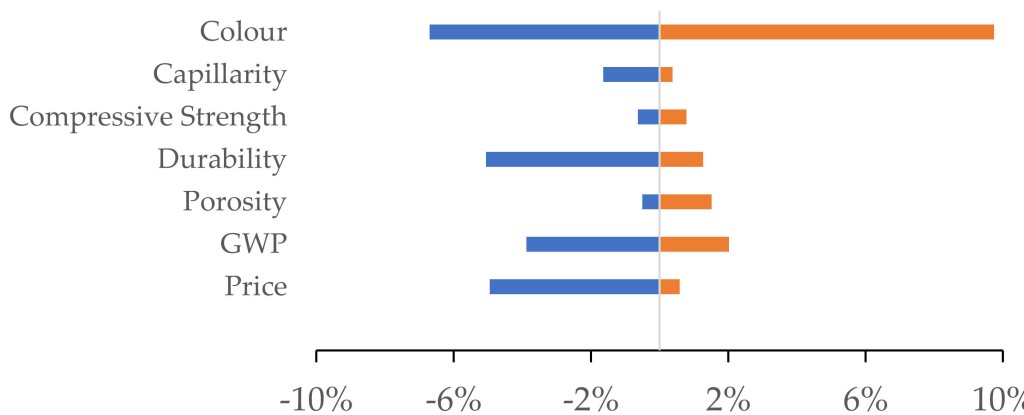

**Figure 6.** Sensitivity analysis of the effect of criteria on the total score. Low and high scenarios correspond to those defined in Table 4.

Additives could be the cause of these great differences between commercial and formulated restoration mortars. Some admixtures and additives can increase the compressive strength. In this study, it was observed that the use of the ground glass additive (G) increases compression strength, as in the case of formulations 23 and 24, with similar composition, but 24 has 10% of glass. The waste glass powder increases the compressive strength of materials and decreases porosity. The obtained results are in line with those of Carsana et al. and Edwards et al. [67,68]. Recent studies show that even a partial substitution of binder by ground glass in lime mortars increases the compressive strength of mortars [69]. The use of natural admixtures, such as brick dust or ground glass mixed with the selected sands increases the durability of mortars, as for example in the salt crystallization test for formulations 24 and 17.

The same methodology was applied to Euville stone, also employed in the Grand Palais applying Equation (14). For Euville stone, we observe that mortars 7 (HSG), 10 (HCSGB), 5 (HCSR), and 9 (HC) seems to be the best options, even better than commercial mortars. When using the weighed score, Equation (15) (Table 12), the best score corresponds to mortar 7 (HSG) followed by mortars 10 (HCSGB), 5 (HCSR), and 9 (HC) due to its properties, cost, GWP, and raw materials availability in Europe. Considering that the building to be restored had two stones, Lutetian and Euville, the mortars that match these two stones are mortars 9 (HC) and 10 (HCSGB). Both mortars are made with natural hydraulic lime number 5.

**Table 12.** Scores of selected mortars for Euville stone in the Royal Palace with weighed coefficients (Equation (15)).

| # | Mortar | Co | CA | CS | D1 | D2 | DE | PO | GW | P | Score |
|---|--------|----|----|----|----|----|----|----|----|---|-------|
| 5 | HCSR | 59 | 100 | 55 | 81 | 86 | 89 | 85 | 61 | 76 | 1019 |
| 7 | HSG | 40 | 88 | 97 | 86 | 89 | 100 | 90 | 60 | 76 | 1047 |
| 9 | HC | 36 | 95 | 61 | 95 | 87 | 83 | 83 | 63 | 77 | 1019 |
| 10 | HCSGB | 30 | 84 | 89 | 98 | 91 | 75 | 85 | 62 | 77 | 1027 |
| 19 | H3.5CS3 | 93 | 92 | 44 | 73 | 84 | 80 | 82 | 64 | 74 | 998 |
| 25 | Lithomex | 30 | 89 | 84 | 85 | 85 | 73 | 87 | 50 | 38 | 929 |
| 26 | Artropierre | 90 | 53 | 84 | 85 | 85 | 63 | 80 | 50 | 38 | 901 |

It is known that natural hydraulic lime is not available everywhere and that many of the materials will differ according to their source. Then, the following assumptions were made to apply this methodology as a selection method for other places in the world:

- Synthetic hydraulic limes (H2 and H3.5) have similar characteristics to natural hydraulic limes;
- The production processes of the raw materials are similar;

- The manufacturing of mortars is similar;
- The physical and chemical properties of the raw materials are similar;
- The distance from the places of origin of the raw materials to the site to be restored must be calculated;
- The cost of raw materials is calculated in Paris, without considering transport costs (better economic studies must be made and this is a perspective of this work).

To apply this methodology to other countries, only materials available in the regions should be considered. For example, there is no production of NHL5 or H5 in Mexico, so these mortars were eliminated from the possibilities in the Maya stone restoration case. In turn, three lots of stones from different sites of the Yucatan Peninsula, were evaluated with this methodology. The first comes from the archaeological site of Dzibichaltun, the second is from the archaeological site of Calakmul, and the third is from a quarry close to the site of Oxpemul. The evaluation was made using the information presented in the Supplementary Material Tables S1, S2, and S8. The mortars highlighted in green do not comply with the color proximity, and those in red were rejected due to having a higher compressive strength than the substrate's or because the binder is unavailable in the region. For the three types of stone, the scores using Equation (14) can be found in the Supplementary Material, Tables S10–S12.

For the case of the first lot of Maya stones, all properties were measured and the comparison with the average of this substrate with the mortars can be seen in the Supplementary Materials, Table S10. It was observed that the impact of color, capillarity, and GWP had a drastic change, since the subtract properties were different and the distance of some of the raw materials were increased. However, in Table 13 it is shown that the best restoration mortar for this type of stone is 24 (H3.5CSGB), followed by the Lithomex commercial mortar.

**Table 13.** Scores for selected mortars for Dzibichaltun site with weighed coefficients (Equation (15)).

| # | Mortar | Co | CA | CS | D1 | D2 | DE | PO | GW | P | Total |
|----|----------|-----|-----|-----|-----|-----|-----|-----|-----|-----|-------|
| 16 | AS2 | 11 | 82 | 30 | 69 | 54 | 28 | 70 | 40 | 56 | 684 |
| 17 | H3.5CS | 4 | 82 | 46 | 41 | 82 | 38 | 72 | 52 | 77 | 751 |
| 23 | H3.5CSB | 14 | 98 | 55 | 79 | 84 | 38 | 77 | 54 | 74 | 887 |
| 24 | H3.5CSGB | 63 | 96 | 68 | 100 | 84 | 34 | 75 | 55 | 75 | 985 |
| 25 | Lithomex | 51 | 76 | 98 | 85 | 85 | 36 | 80 | 50 | 38 | 895 |
| 26 | Artropierre | 0 | 88 | 98 | 85 | 85 | 26 | 73 | 50 | 38 | 851 |

Considering that the method is modifiable, in the case of the second batch of Mayan stones, the durability of freezing and thawing was not considered, since this site does not present temperatures below 14 °C. Density variations in more than 20 samples ranged from 1520 to 2315 kg/m$^3$; hence, due to the uncertainty of these measurements, it was decided to take density out of the calculation. In the Supplementary material, Table S11 shows the total calculation of the scores for the restoration mortars simulating the restoration of the stone at Calakmul. Table 14 shows the weighed scores for the selected mortars for the Calakmul site (Equation (15)). Only the best three are displayed. According to this methodology, mortar 23 (H3.5CSB) is the best option for the restoration of the Calakmul stone.

**Table 14.** Scores for selected mortars for the Calakmul stone with weighed coefficients (Equation (15)).

| # | Mortar | Co | CA | CS | D1 | PO | GW | P | Total |
|----|---------|-----|-----|-----|-----|-----|-----|-----|-------|
| 16 | AS2 | 0 | 82 | 67 | 69 | 83 | 28 | 56 | 564 |
| 22 | H3.5CSG | 0 | 97 | 91 | 84 | 91 | 38 | 73 | 692 |
| 23 | H3.5CSB | 0 | 98 | 92 | 79 | 90 | 38 | 74 | 685 |

For the case of the third batch of stones, the total calculation of the assignment values for the restoration mortars simulating the need to restore the stone of Oxpemul is presented

in the Supplementary material, Table S12, and in Table 15, using weighing coefficients. The result was that mortar 24 (H3.5CSGB) is the best option, followed by the commercial mortar Lithomex, Artropierre, and mortar 23 (H3.5CSB).

**Table 15.** Scores for selected mortars for the Oxpemul stone with weighed coefficients (Equation (15)).

| # | Mortar | Co | CA | CS | D1 | D2 | DE | PO | GW | P | Total |
|---|--------|----|----|----|----|----|----|----|----|---|-------|
| 13 | ACSGB | 0 | 87 | 0 | 71 | 82 | 95 | 95 | 41 | 66 | 818 |
| 17 | H3.5CS | 3 | 82 | 1 | 41 | 82 | 83 | 98 | 35 | 77 | 741 |
| 23 | H3.5CSB | 11 | 98 | 10 | 79 | 84 | 83 | 93 | 35 | 74 | 864 |
| 24 | H3.5CSGB | 51 | 96 | 23 | 100 | 84 | 87 | 95 | 38 | 75 | 967 |
| 25 | Lithomex | 52 | 76 | 89 | 85 | 85 | 85 | 90 | 50 | 38 | 946 |
| 26 | Artropierre | 0 | 88 | 89 | 85 | 85 | 95 | 97 | 50 | 38 | 934 |

Indeed, the best options for restoring these stones from Mayan regions have been products with hydraulic lime. If these are not available in large quantities, mortars with aerial lime that also give good results are formulations 11 (AS), 13 (ACSGB), and 16 (AS2).

This method proposal is the first attempt in the field, and further studies are needed. The real application of this method is limited to choosing the best options from a sustainable point of view, which will have to be analyzed in other properties, such as adhesion and possibly modified with additives to be used in cultural heritages. The difference with other methods is the consideration of costs, GWP, or durability of the materials.

This method needs the characteristics of the substrates and mortars to be applied. In the short term, more properties can be added, which leaves an open line of research in the search for a global method for choosing sustainable materials.

This method could be applied to other construction materials with the modifications of the properties to be measured. The limitations of this method are the properties proposed since, depending on the specific work of a material, certain properties may be crucial for its use.

## 4. Conclusions

The color differences between the substrates and the mortars can be very noticeable, so it is essential to first define the tone, to know the exact composition and amounts of pigments, before continuing with the mortar selection processes, as this will impact other indicators in the method, such as GWP.

The durability of a mortar is an essential factor regarding its useful life. The salt crystallization test is more destructive than the freeze-thaw test. There are mortars that, depending on their chemical composition or their physical properties, may be more susceptible to one test or another. In the mortars presented in this work, the mortars with hydraulic lime presented better durability than those with air lime.

Recyclable additives/admixtures will improve some mortar properties and reduce the GWP. The equation presented in this work allows for a simplified GWP estimation that does not require new Life Cycle Assessment.

This work presents a methodology for selecting restoration mortars adaptable for different buildings (material, environments). The methodology is based on the properties of materials and substrates. According to the circular economy principles, it allows a fast, easy, and efficient procedure to select adequate restoration mortars fulfilling sustainability requirements from an environmental, technological, and commercial point of view. The equations proposed for the total score shows a coherent sensitivity analysis with less than 10% participation for each property.

This selection methodology can be modified depending on expert judgment, allowing for the inclusion of additional properties not included in this paper, such as adhesion or hydric properties. It can also be used as an iterative approach of eco-design of mortars with additives. Each building has its own particularities, not only concerning employed materials and environmental conditions but also cultural values to be preserved, which will

require particular restitution methods. This methodology can be used to compare different mortars or construction materials. In each case, it is essential to consider the national or regional regulations.

Using the selection methodology, a suitable mortar was identified for the restoration of the Royal Palace of Paris, built with Lutetian stone and the Euville stone, in this case, formulation 10 (HCSGB). While further analyses may be needed to validate this conclusion, this mortar is the best starting point.

Regarding the simulation for stones from the Maya region, it is concluded that mortars elaborated with hydraulic lime 3.5 are optimal for restoration, especially formulation 24 (H3.5CSGB), except for the Calakmul stone, which has a lower compressive strength; in that case, mortar 23 (H3.5CSB) is a better choice.

**Supplementary Materials:** The following supporting information can be downloaded at: https://www.mdpi.com/article/10.3390/geosciences12100362/s1, Figure S1: Durability of restoration mortars in salt crystallization test; Figure S2: Durability of restoration mortars in freezing thawing test; Table S1: Colour of Mortars in green the mortars with less than 10 and in blue commercial mortars; Table S2: Physical and mechanical properties of limestones; Table S3: Mass difference in the Durability of restoration mortars after the last cycle; Table S4: Emission factors of inputs; Table S5: LCA calculation of a restoration mortar; Table S6: Additives normally use in mortars; Table S7: XRAY composition of raw materials and admixtures; Table S8: Distance of the raw materials for Mayan monuments; Table S9: LCA of mortars for Different sites; Table S10: Assignment of values in restoration mortars for Dzibichaltun stone; Table S11: Assignment of values in restoration mortars for Calakmul; Table S12: Assignment of values in restoration mortars for Oxpemul stones [70–96].

**Author Contributions:** J.D.-B.: Conceptualization, methodology, software, original draft preparation, writing—review and editing; B.M.: Supervision, formal analysis, review, and editing; J.R.: formal analysis, review, and editing; J.C.S.R.: Validation, formal analysis, review, and editing. All authors have read and agreed to the published version of the manuscript.

**Funding:** This research was funded by the programs "Make Our Planet Great Again" and "Initiative d'excellence Paris Seine. This paper is supported by the European Union's Horizon 2020 research and innovation program under grant agreement No 101007531 SCORE (Sustainable COnservation and REstoration of built cultural heritage).

**Data Availability Statement:** Not applicable.

**Acknowledgments:** The authors thank the programs "SORE", "Make Our Planet Great Again" and "Initiative d'excellence Paris Seine" for providing financial support for the development of this research. Socli, Rocamat, Briqueterie d'Allone and Fédération du Verre for providing the information and materials for this work.

**Conflicts of Interest:** The authors declare no conflict of interest.

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
