# Peer review of "A Selection Method for Restoration Mortars Using Sustainability and Compatibility Criteria"

_geosciences, doi:10.3390/geosciences12100362_

Round 1

Reviewer 1 Report

The topic is of interest and the text is well written. The authors present a methodology to check the best lime mortars for each restoration purpose. They consider various properties such as mechanical compatibility, economic issues, Global Warning Potential and aesthetic ones, among others. Most of them are of absolute coherence and also of significant novelty.

I propose correcting some typos and clarfying some test procedures as indicated in the attached file.

Reviewer 2 Report

The manuscript entitled “Selection method of restoration mortars using sustainability and compatibility criteria” by J. Diaz-Basteris and coauthors presents an original method for quantitatively assessing the properties of restoration mortars.

However, there are some points that must be addressed before publication to improve the manuscript quality:

General comment

The paper proposes a sound methodology, attempting to find universal compatibility criteria that are strongly needed in the conservation field. However, some critical points to the proposed method must be highlighted by the authors in the introduction and, possibly, in the conclusive remarks. In particular, the authors consider their case as a general condition for any restoration mortar, but several variables must affect the final results. Different types of mortars exist depending on their function. For instance, mortars for injections or consolidation, have different properties in their fresh state. In addition, application methods also affect the performances of mortars, as well as the state of conservation of the substrate.

Of course, it would be a huge and long work to find a universal method for the formulation of all types of restoration mortars, but the authors should specify in their paper what the limitations are and that the presented method is one of the first attempts in the field and further studies are needed.

A discussion must be added for the interpretation of the results. You might rename Section 2 as “Results and Discussion”.

Detailed comments

Introduction

Lines 24-26: The definition of mortar you provided lacks the term “admixture”. The standard EN 16572 defines a mortar as a “material traditionally composed of one or more (usually inorganic) binders, aggregates, water, possible additives, and admixtures combined to form a paste used in masonry for bedding, jointing and bonding, and for surface finishing (plastering and rendering) of masonry units, which subsequently sets to form a stiff material”. Just you, in fact, have used the term “admixture” in the first paragraph of section 2. Materials and Methods.

Lines 26-29: As far as the function of restoration mortar is concerned, you should consider also listing the archaeological contexts and materials (e.g., mosaics, mural painting, etc.). You may consider these readings on the subject:

Miriello, D., et al. (2021). Hydraulicity of lime plasters from Teotihuacan, Mexico: a microchemical and microphysical approach. Journal of Archaeological Science, 133, article number 105453. https://doi.org/10.1016/j.jas.2021.105453

Miriello, D., et al. (2018). New compositional data on ancient mortars and plasters from Pompeii (Campania – Southern Italy): Archaeometric results and considerations about their time evolution. Materials Characterization, 146, 189–203. https://doi.org/10.1016/j.matchar.2018.09.046

Rispoli, C., et al. (2020). Unveiling the secrets of Roman craftsmanship: mortars from Piscina Mirabilis (Campi Flegrei, Italy). Archaeological and Anthropological Sciences, 12, article number 8, https://doi.org/10.1007/s12520-019-00964-8

Rispoli, C., et al. (2019). The ancient pozzolanic mortars of the thermal complex of Baia (Campi Flegrei, Italy). Journal of Cultural Heritage. https://doi.org/10.1016/j.culher.2019.05.010

Lines 30-31: You only considered ancient stone as a substrate to apply restoration mortar, but ancient structures are also composed of bricks and ancient mortars.

Line 50: You may consider citing the following paper for composite material mortar/substrate:

Cultrone, G., et al. (2007). Durability of masonry systems: A laboratory study. Construction and Building Materials, 21, 40-51. https://doi.org/10.1016/j.conbuildmat.2005.07.008

Materials and Methods

More geological information and references are needed on the stones (Lutetian, Euville, Maya) used as substrates.

Appendix

From pages 24 to 34: all annexes are named in French. Please correct them.

END OF THE REVIEW
